# The role of chromatin dynamics under global warming response in the symbiotic coral model Aiptasia

Eviatar Weizman[1] & Oren Levy[1]

Extreme weather events frequency and scale are altered due to climate change. Symbiosis between corals and their endosymbiotic-dinoflagellates (*Symbiodinium*) is susceptible to these events and can lead to what is known as bleaching. However, there is evidence for coral adaptive plasticity in the role of epigenetic that have acclimated to high-temperature environments. We have implemented ATAC-seq and RNA-seq to study the cnidarian-dinoflagellate model *Exaptasia pallida* (Aiptasia) and expose the role of chromatin-dynamics in response to thermal-stress. We have identified 1309 genomic sites that change their accessibility in response to thermal changes. Moreover, apo-symbiotic Aiptasia accessible sites were enriched with *NFAT*, *ATF4*, *GATA3*, *SOX14*, and *PAX3* motifs and expressed genes related to immunological pathways. Symbiotic Aiptasia accessible sites were enriched with *NKx3-1*, *HNF4A*, *IRF4* motifs and expressed genes related to oxidative-stress pathways. Our work opens a new path towards understanding thermal-stress gene regulation in association with gene activity and chromatin-dynamics.

[1] Mina and Everard Goodman Faculty of Life Sciences, Bar-Ilan University, Ramat Gan 52900, Israel. Correspondence and requests for materials should be addressed to E.W. (email: eviatar.weizman@gmail.com) or to O.L. (email: oren.levy@biu.ac.il)

Coral reefs are one of the most diverse and important marine ecosystems, providing a home to hundreds of thousands of species[1], including almost a third of the world's marine fish species[2]. Coral reefs support more species per unit area than any other marine ecosystem, making them an important reservoir for biological diversity and complexity. About 15% of the world's population lives within 100 km of coral reef ecosystems, with many people depending on coral reefs for their livelihood. The importance of coral reefs for tourism, fishing, building materials, coastal protection, and drug discovery cannot be underestimated.

Corals live in close association with a variety of eukaryotic and prokaryotic micro-organisms, potentially providing additional adaptive capacity to the holobiont. The most studied and well-known endosymbiosis between corals and their dinoflagellate algae (family Symbiodiniaceae) has a profound contribution to coral reefs rapid ecological success over geological history[3]. Over the past 200 million years corals have had a crucial role in shaping tropical oceans, but yet they appear to be highly vulnerable to environmental stress in general, and specifically to anthropogenic factors including climate change[4]. Over the past several decades, reefs throughout the world have been affected by local anthropogenic stressors and climate change—as much as 75% of the world's coral reefs are threatened and as many as 95% may be in danger of being lost by mid-century[5]. This can be attributed to mass bleaching events that are tied to global warming[6,7], but local stressors associated with overharvesting and coastal development (urban and agricultural) are also major contributors to this global decline[8]. Bleaching occurs when the coral host loses its symbiotic-partners, the dinoflagellate Symbiodinium[9]. The loss of these symbionts can have adverse effects on the health of corals and eventually lead to their death.

Demystifying the fundamental mechanisms of symbiont presence or absence is important for future conservation efforts and especially for attempts of assisted evolution[10]. Today enormous efforts are made involving transcriptomics studies to reveal the potential mechanism of coral adaptation to a variable environment and stressors[11,12]. At present there are only a few studies trying to unravel epigenetic mechanisms in cnidarians including DNA methylation in corals[13,14]; and histone markers like in the brackish sea anemone Nematostella[15]. However, still a major knowledge gap remains, especially in histone modifications, and chromatin dynamics, regarding the regulatory and epigenetic mechanisms controlling these cellular and molecular pathways in corals[16,17], partially due to the fact that in corals DNA methylation levels correlate broadly and uniformly with expressed 'housekeeping' genes, whereas genes responsible for inducible or cell-specific functions are weakly methylated[18].

Eukaryotic DNA is wound around histone proteins in a complex called nucleosome. This complex is vastly regulated as histones are removed to expose regulatory sites, such as cis-regulatory elements (CREs) and promoters, to allow binding of transcription factors and other regulatory proteins. Identification of enriched motifs within these active CREs can, therefore, reveal genes associated with a transcriptional regulatory network[19]. Genome-wide mapping of transcription factors binding to chromatin is frequently done by chromatin immunoprecipitation (ChIP) based methods, such as ChIP-seq[20]. Conversely, these techniques are expensive and require a significant amount of tissue and extensive processing of the sample. The Assay for Transposase-Accessible Chromatin with high-throughput sequencing (ATAC-seq) is a novel technology that favors the sequencing of accessible chromatin loci[21] and holds promise to overcome these limitations. While ATAC-seq is a powerful and promising approach for epigenetic regulation research, it is relatively new and has primarily been applied within well-characterized model systems.

In this study, we aim to elaborate an understanding of cnidarian gene expression and regulation by revealing the interplay between chromatin accessibility aligned with gene expression dynamics, and its relation to the symbiotic state of the animal under temperature elevation reflecting global warming. For this goal, we choose to work with the symbiotic sea anemones Aiptasia pallida (sensu Exaiptasia pallida). The relationship with Symbiodinium is facultative, making this a convenient laboratory surrogate for studying coral and other cnidarians symbiosis[22]. We optimized an ATAC-seq protocol to detect accessible chromatin regions in the symbiotic and apo-symbiotic morph of Aiptasia gradually introduced to thermal stress. By integrating chromatin accessibility profiles with transcription profiles (RNA-seq) we revealed 1309 genomic sites that change their accessibility in response to thermal changes. Moreover, apo-symbiotic Aiptasia accessible sites were enriched with NFAT, ATF4, GATA3, SOX14 and PAX3 motifs and expressed genes related to immunological pathways. Symbiotic Aiptasia accessible sites were enriched with NKx3-1, HNF4A, IRF4 motifs and expressed genes related to oxidative-stress pathways. This work opens a new avenue towards understanding thermal stress gene regulation with the association of gene activity and chromatin accessibility. The work presented here shows that chromatin structure may act as a mechanism for adaptive response in regulating gene expression in cnidarian symbioses under global warming.

## Results

**Chromatin accessibility profiles of Aiptasia.** To sensitively measure high-resolution chromatin accessibility at different symbiotic states of Aiptasia in response to heat-stress, we used the Assay for Transposase Accessible Chromatin using sequencing (ATAC-seq). We optimized the ATAC-seq protocol[21,23] for Aiptasia by including a step of symbiotic algae removal and native nuclei isolation by mechanical homogenization before the transposition step (see methods). We generated and sequenced ATAC-seq libraries, as well as input control, to a median depth of 8 million unique, high-quality mapping reads per sample. The experimental design included a total of four groups, in which two were introduced to heat-stress and two to constant temperature conditions serving as the control, over a period of 28 days. This was done for both symbiotic (symbiosis with Breviolum (i.e., clade B)[24]) and apo-symbiotic state. The two heat-stress groups were exposed to gradually rising temperatures (+0.5 °C per day to a max temperature of 34 °C) and sampled at four designated temperature steps (n = 3 per time point) as shown in Supplementary Table 1. The photosynthetic yield (Fv/Fm) was measured along with the experiment as an indicator of symbiosis state as presented in Fig. 1. Control groups that were held under constant temperature showed no significant change in photosynthetic yield, while the symbiotic Aiptasia groups that were introduced to stress showed a significant change (One-way ANOVA: P-value < 0.001) in photosynthetic yield in response to the stress (symbiotic control Fv/Fm was at the rage of 0.6–0.67 and treated symbiotic anemones Fv/Fm was 0.25 at the end of the experiment) (Fig. 1b). The apo-symbiotic Aiptasia photosynthetic yields were measured, including algae count in addition (no detectable algae were notified), to ensure their apo-symbiotic state (see Fig. 1a, c). Morphological changes, mainly the body expansion of Aiptasia, were observed during the experiment between day 21 (34 °C) and after 1 week at this temperature (day 28 of the experiment, temperature 34 °C). The anemones showed more contraction behavior (both tentacles and body) indicating a high-stress state[25].

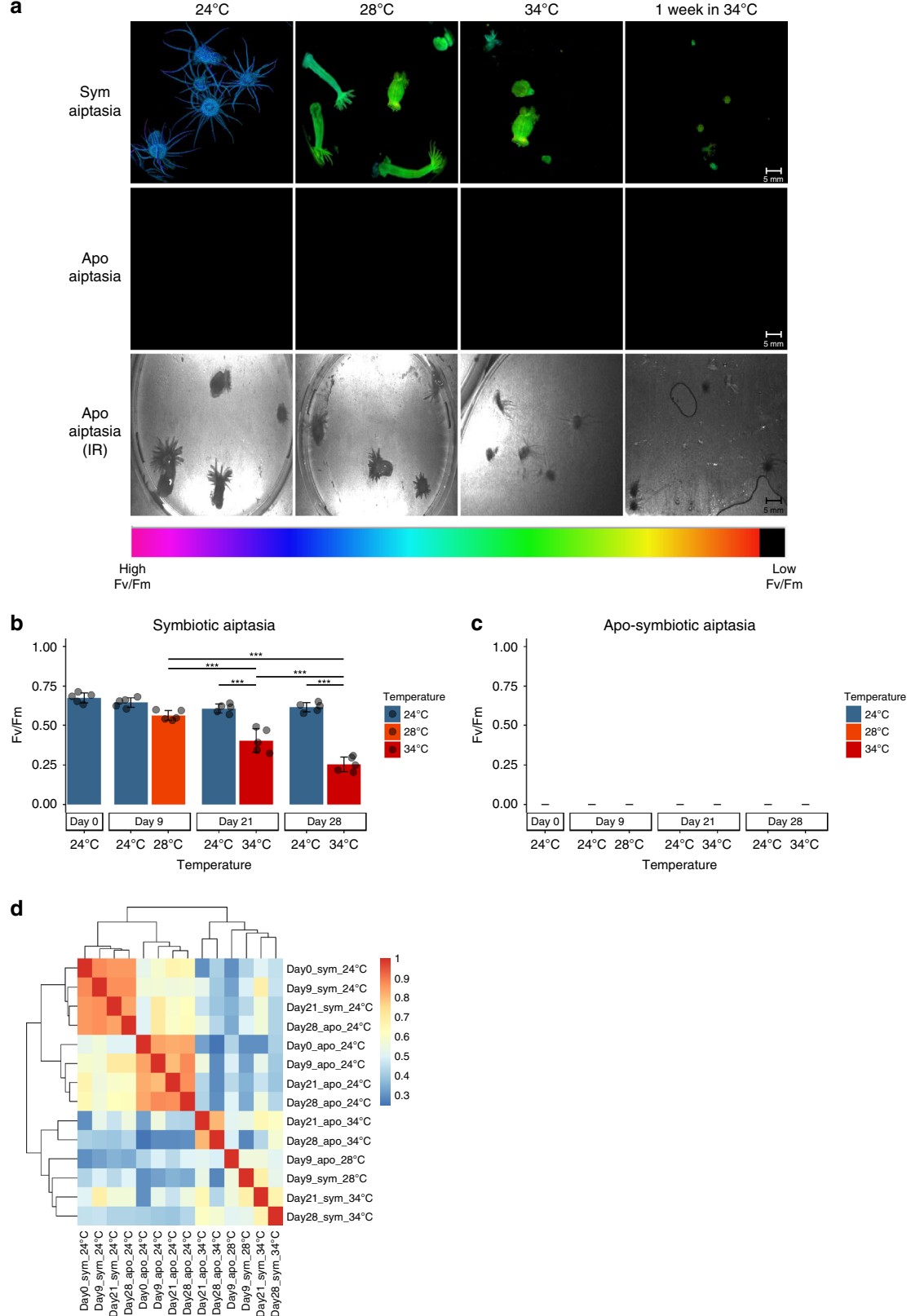

**Fig. 1** IPAM Fv/Fm measurements of symbiotic and apo-symbiotic Aiptasia prior to sampling for symbiosis state validation: **a** IPAM image as captured prior to sampling color scale indicates Fv/Fm value, and infra-red (IR) image indicates the apo-symbiotic Aiptasia position in the frame. **b** Quantitative results of Fv/Fm as measured with IPAM for symbiotic Aiptasia $n = 5$, One-way ANOVA: $P$-value $< 0.001$, Post-hoc Tukey-HSD: $P$-value $< 0.001$. **c** Quantitative results of Fv/Fm as measured with IPAM for apo-symbiotic Aiptasia $n = 5$. **d** ATAC-seq signal within consensus ATAC-seq peaks was compared between all samples using Spearman's ρ to cluster samples

**Table 1 Correlation between ATAC-seq biological replicates**

| Group[a] | rep 1 vs. rep 2[b,c] | rep 1 vs. rep 3[b,c] | rep 2 vs. rep 3[b,c] | Average $R^2$ | SD |
|---|---|---|---|---|---|
| Day0_apo_24 °C | 0.815 | 0.872 | 0.827 | 0.838 | 0.025 |
| Day9_apo_24 °C | 0.864 | 0.798 | 0.857 | 0.840 | 0.030 |
| Day21_apo_24 °C | 0.829 | 0.672 | 0.833 | 0.778 | 0.075 |
| Day28_apo_24 °C | 0.827 | 0.869 | 0.882 | 0.859 | 0.023 |
| Day0_sym_24 °C | 0.952 | 0.820 | 0.838 | 0.870 | 0.058 |
| Day9_sym_24 °C | 0.969 | 0.977 | 0.968 | 0.971 | 0.004 |
| Day21_sym_24 °C | 0.966 | 0.949 | 0.922 | 0.946 | 0.018 |
| Day28_sym_24 °C | 0.860 | 0.934 | 0.882 | 0.892 | 0.031 |
| Day9_apo_28 °C | 0.934 | 0.942 | 0.906 | 0.927 | 0.015 |
| Day9_sym_28 °C | 0.978 | 0.993 | 0.996 | 0.989 | 0.008 |
| Day21_apo_34 °C | 0.797 | 0.883 | 0.720 | 0.800 | 0.067 |
| Day28_apo_34 °C | 0.700 | 0.738 | 0.966 | 0.801 | 0.117 |
| Day21_sym_34 °C | 0.676 | 0.938 | 0.698 | 0.771 | 0.119 |
| Day28_sym_34 °C | 0.661 | 0.952 | 0.659 | 0.757 | 0.138 |

[a]Group name format: sampling day/symbiosis state/temperature
[b]Pearson product-moment correlation coefficient
[c]$P$-value < 0.01 for all results

As presented in Table 1, the biological replicates ATAC-seq libraries were highly similar (average adj. $R^2 = \sim0.86$ with SD = 0.073 and $P$-value < 0.001), demonstrating highly reproducible data from Aiptasia, whole animal sampling. The ability to cluster samples by their stress stage also shows that chromatin accessibility is strikingly different between the two morphs during the response to heat stress (Fig. 1d). These differences between heat-stress stages are likely due to changes in accessibility within host cell populations in response to heat and symbiotic state. Furthermore, the significant peaks (36,999–83,061 peaks with $-\log(p\text{-value}) > 5$) from all test groups were clustered around transcriptional start sites (TSSs, see Supplementary Fig. 1). Together, these results indicate that reproducible high-resolution chromatin accessibility can be obtained from small amounts (at least an order of magnitude less than standard ChIP-seq or DNase-seq) of complex, whole animal Aiptasia samples.

**A glimpse into Aiptasia genome regulatory regions**. ATAC-seq favors accessible sites of chromatin and thus, we expected to find more regulatory related genomic features enriched within the sequenced libraries[19]. Indeed, the libraries showed on average that Promoters-TSS (defined as 1500 bp upstream to 400 bp downstream from the first nucleotide of the gene) features were enriched by over 40%, as shown in Fig. 2a, b. In addition, many open chromatin regions identified by ATAC-seq were within proximal (5000 bp-1501 bp upstream from the first nucleotide of the gene) or distal (>5000 bp downstream from the first nucleotide of the gene) intergenic regions, suggesting these regions may act as distant regulatory elements. Moreover, the accessible landscape between the symbiotic to the apo-symbiotic sea anemones was different (20–40% difference), suggesting the immense influence of symbiont presence within the anemone tissues, as has been shown before by our group[26] - Fig. 2c–f. We have observed that differences in the regulatory landscape may appear already in the basal condition (Fig. 2c), and by that, we can assume that they are dependent on symbiont presence. We performed a motif enrichment analysis flowed by GO enrichment of biological processes (Supplementary Fig. 2). We have found that under basal conditions, symbiotic morphs top pathways were related to symbiosis establishment and maintenance (Positive regulation of transport, modulation by host of genome replication, regulation of symbiosis). The apo-symbiotic morphs, however, showed enriched pathways related to organism homeostasis (Positive

regulation of mRNA catabolic process, regulation of sphingolipid mediated signaling pathway, TRIF-dependent toll-like receptor signaling pathway).

**The regulatory landscape of DNA dynamics during heat-stress**. The analysis of the regulatory DNA landscape under heat-stress response was mapped displaying genomic sites that change in response to the rising temperature of both symbiotic and apo-symbiotic sea anemones. We identified 853 heat responsive sites in apo-symbiotic Aiptasia, which clustered into five accessibility patterns (Fig. 3a, b), while 787 heat responsive sites in symbiotic Aiptasia clustered into four accessibility patterns (Fig. 4a, b). Between the two morph-types only 331 overlapping sites were identified varying in response to rising temperature, implying different response pathways taken by each morph. Moreover, many sites within the clusters resided in proximity to genes previously associated with heat response (Figs 3c and 4c). For example, in apo-symbiotic Aiptasia the site overlying the promoter of AP_hsp90.a1 was identified (Fig. 3c), while in symbiotic Aiptasia the site overlying the promoter of AP_hspa12a (Heat Shock Protein Family A (Hsp70) Member 12 A) was recognized (Fig. 4c). Heat Shock Proteins are molecular chaperones that regulate protein structure and function during and after stress, including heat-stress[27,28]. To further explore the transcription factors mediating the response to heat in association to accessible sites found, we performed motif enrichment analysis among clusters (Figs 3d and 4d). In apo-symbiotic Aiptasia it was found that *NFAT* motifs were enriched within cluster IV sites. In symbiotic Aiptasia SMAD motifs were enriched within clusters I, II and III, which cover all temperature ranges of this experiment (Fig. 4b, d). Further examination of transcription factors expression with RNA-seq showed that expression levels were correlated with extended promoter (0–1500 bp downstream to TSS) accessibility of each enriched transcription factors (Fig. 5). Remarkably, the expression changes of transcription factors were indicative of the expression profiles in the respective clusters, further supporting an association of these transcription factors to the heat stress response.

**Functional pathways analysis reveals response to heat-stress**. The canonical pathways enrichment analysis was performed to identify differences in morph response to heat stress. Thus, we chose the top pathways (by $-\log_{10}$ (P-value)) enriched for each

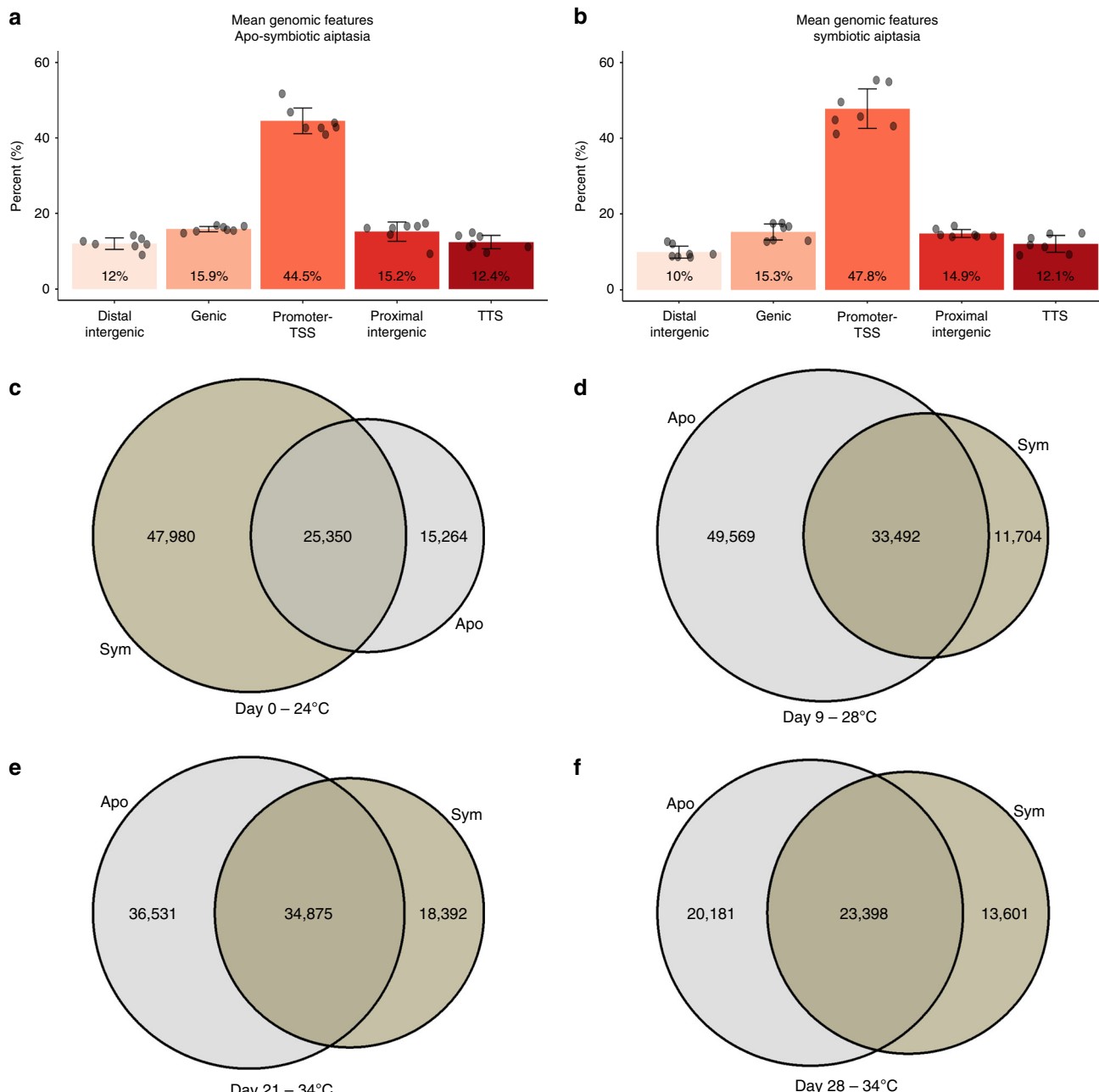

**Fig. 2** Genomic features mean distribution across ATAC-seq libraries. **a** The average percentage. of genomic features of Apo-symbiotic Aiptasia calculated from replicates from all groups and sampling points ($n = 7$). **b** The average percentage of genomic features of symbiotic Aiptasia calculated from replicates from all groups and sampling points ($n = 7$). **c–f** Venn diagrams of overlap of apo-symbiotic Aiptasia (gray) and symbiotic Aiptasia (brown-green) ATAC-seq peaks at different temperatures

morph and paralleled the results of the two groups (Fig. 6). In the symbiotic morph, samples were more enriched with pathways related to oxidative stress such as RAR activation ($-\log_{10}$ (P-value) >2 at day 21 and day 28), Oxidative stress ($-\log_{10}$ (P-value) >3.6 at day 21 and day 28), NRF2-mediated Oxidative Stress Response ($-\log_{10}$ (P-value) >2.46 at day 21 and day 28) and Genes upregulated in response to oxidative stress ($-\log_{10}$ (P-value) >3 at day 28). While the apo-symbiotic morph samples were more enriched with pathways related to immune response such as *TGF-β* Signaling ($-\log_{10}$ (P-value) >3.9 at day 21 and day 28), *NF-kB* signaling ($-\log_{10}$(P-value) >1.45 at day 21 and day 28) and p53 signaling pathway ($-\log_{10}$ (P-value) >1.69 at day 21 and day 28).

## Discussion

Expression patterns of many protein-coding genes are orchestrated in response to exogenous stimuli, as well as cell-type-specific developmental programs. It has been shown that dynamic chromatin movements and interactions in the nucleus play a crucial role in gene regulation[29]. To illuminate the chromatin dynamic rearrangement in response to rising temperatures with relation to the presence or absence of the symbiotic algae in *A. pallida*, we undertook comprehensive analysis using ATAC-seq and RNA-seq data obtained from one-month of experiment. The experiment was designed to subject both apo-symbiotic Aiptasia and symbiotic Aiptasia to gradually rising temperature as shown in Supplementary Table 1. On the outside and during the

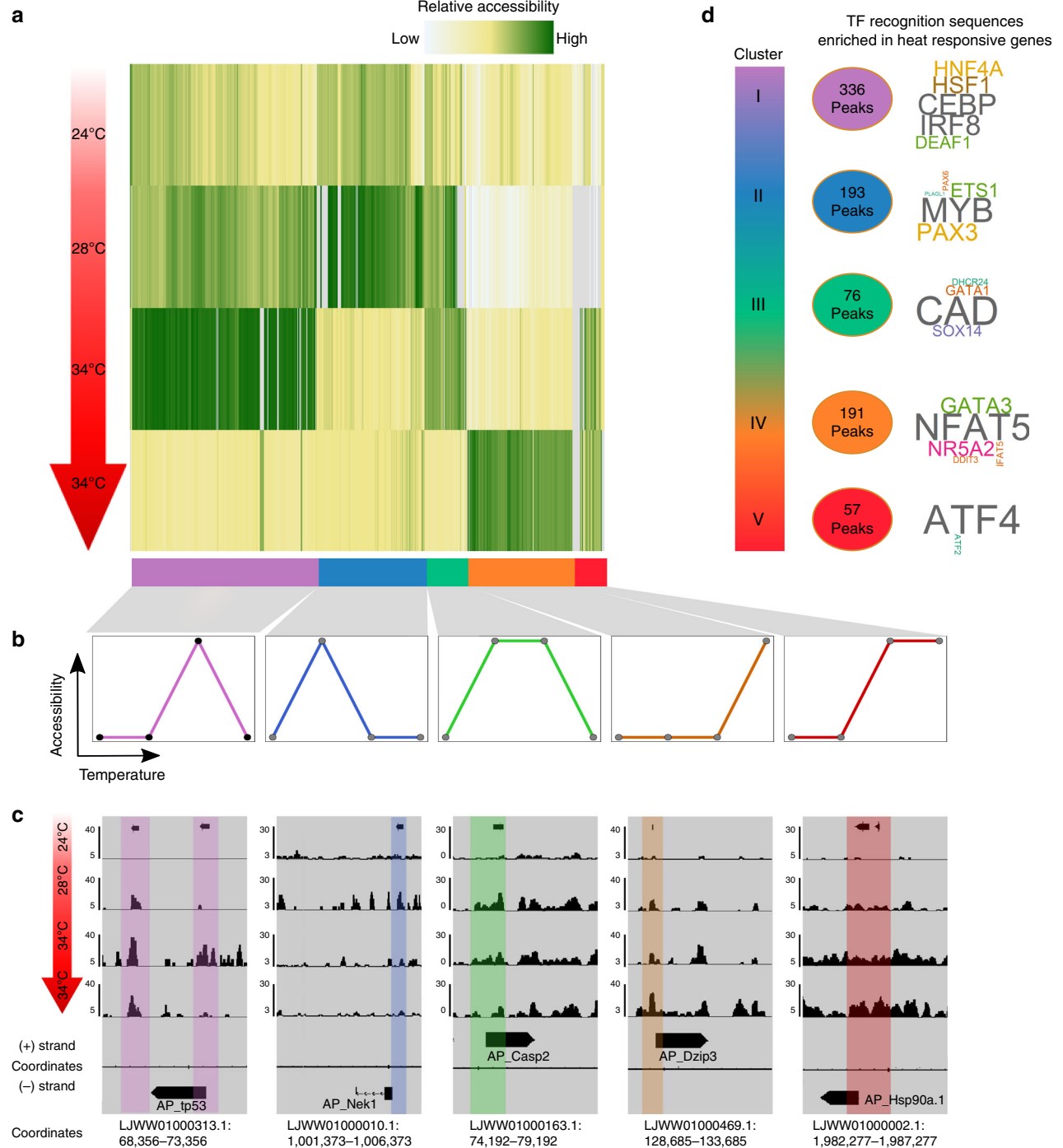

**Fig. 3** Dynamic chromatin changes during heat-stress in apo-symbiotic Aiptasia. **a** 853 ATAC-seq sites identified for each temperature were clustered, yielding five accessibility clusters: (I, purple) sites accessible at day 21 in 34 °C, (II, blue) accessible at day 9 in 28 °C, (III, green) accessible at day 9 in 28 °C and day 21 in 34 °C, (IV, orange) accessible at day 28 in 34 °C, and (V, red) accessible at day 21 and day 28 in 34 °C. **b** Characteristic patterns of ATAC-seq sites accessibility. **c** Representative examples of genes from each accessibility clusters. Each window is 5 kb; specific coordinates are mentioned bellow tracks; sites of interest are highlighted in a matching color. **d** HOMER motif enrichment relative to background within each cluster is represented as a word cloud

experiment, changes were distinguised in size and morphology of both morphs types after reaching a temperature of 34 °C and deteriorated during the following days at the same high temperature as expected[25] (these changes are temperature related as salinity and water quality were kept stable).

To access chromatin level changes, we measured accessible sites using ATAC-seq. We could identify 36,999–83,061 peaks with −log(p-value) >5, enabling the prediction of transcription

factors binding sites within the accessible genome and specifically identifying the sites that changed in response to rising water temperature. As shown in Fig. 2, regulatory sites, such as promoters and proximal intergenic regions, etc., were more represented in our data sets relative to their portion within the genome. Comparison of data peaks between morphs highlighted stark differences throughout the experiment (Fig. 2c–f), suggesting the different physiological and metabolic requirements and survival

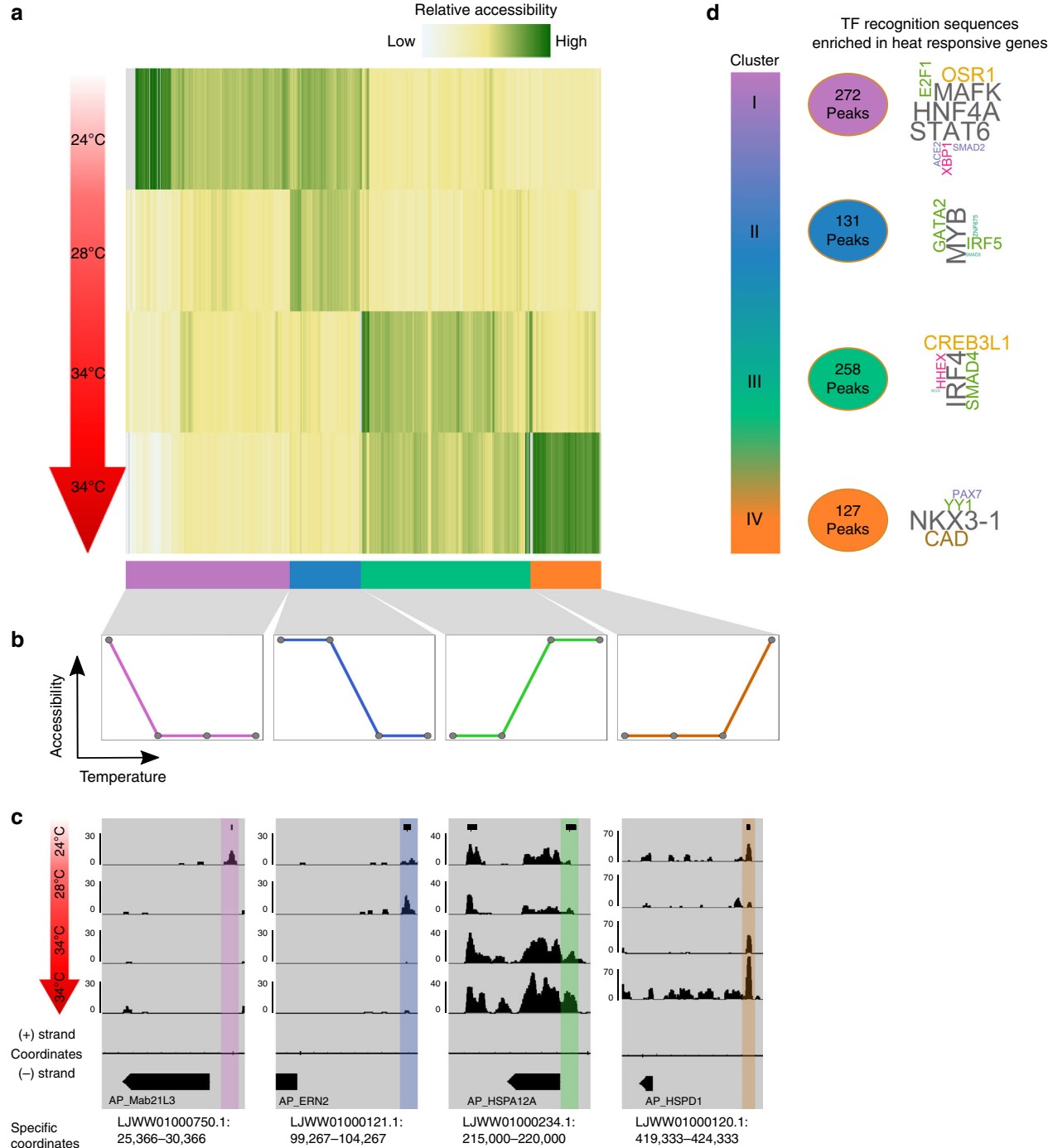

**Fig. 4** Dynamic chromatin changes during heat-stress in symbiotic Aiptasia. **a** 787 ATAC-seq sites identified for each temperature were clustered, yielding four accessibility clusters: (I, purple) sites accessible at day 0 in 24 °C, (II, blue) accessible at day 0 in 24 °C and day 9 in 28 °C, (III, green) accessible at day 21 and day 28 in 34 °C, (IV, orange) accessible at day 28 in 34 °C. **b** Characteristic patterns of ATAC-seq sites accessibility. **c** Representative examples of genes from each accessibility clusters. Each window is 5 kb; specific coordinates are mentioned bellow tracks; sites of interest are highlighted in a matching color. **d** HOMER motif enrichment relative to background within each cluster is represented as a word cloud

strategies used, depend on symbiont presence within the host tissue. In basal condition the two morphs invest their energy differently while apo-symbiotic Aiptasia maintains homeostasis, the symbiotic Aiptasia invest in symbiotic preservation (Supplementary Fig. 2), this starting point farther affects the two morphs heat response profile. With that observation in mind, we have found sites that significantly change their accessibility in response to heat (P-value < 0.05); around 800 sites changed their accessibility throughout the experiment, with only 331 overlapping sites

between morphs (Figs 3 and 4). Those sites were clustered by patterns (5 clusters for apo-symbiotic morph and 4 clusters for symbiotic morph) and motif enrichment analysis revealed different motifs associated with each morph and clusters. The low overlapping rate between the morphs accessible sites means a different profile response to heat stress. Between the morphs, apo-symbiotic clusters 5 and 4 (in Fig. 3) present the same pattern as symbiotic clusters 3 and 4 respectively (in Fig. 4), although different sites and genes participate in those groups. That

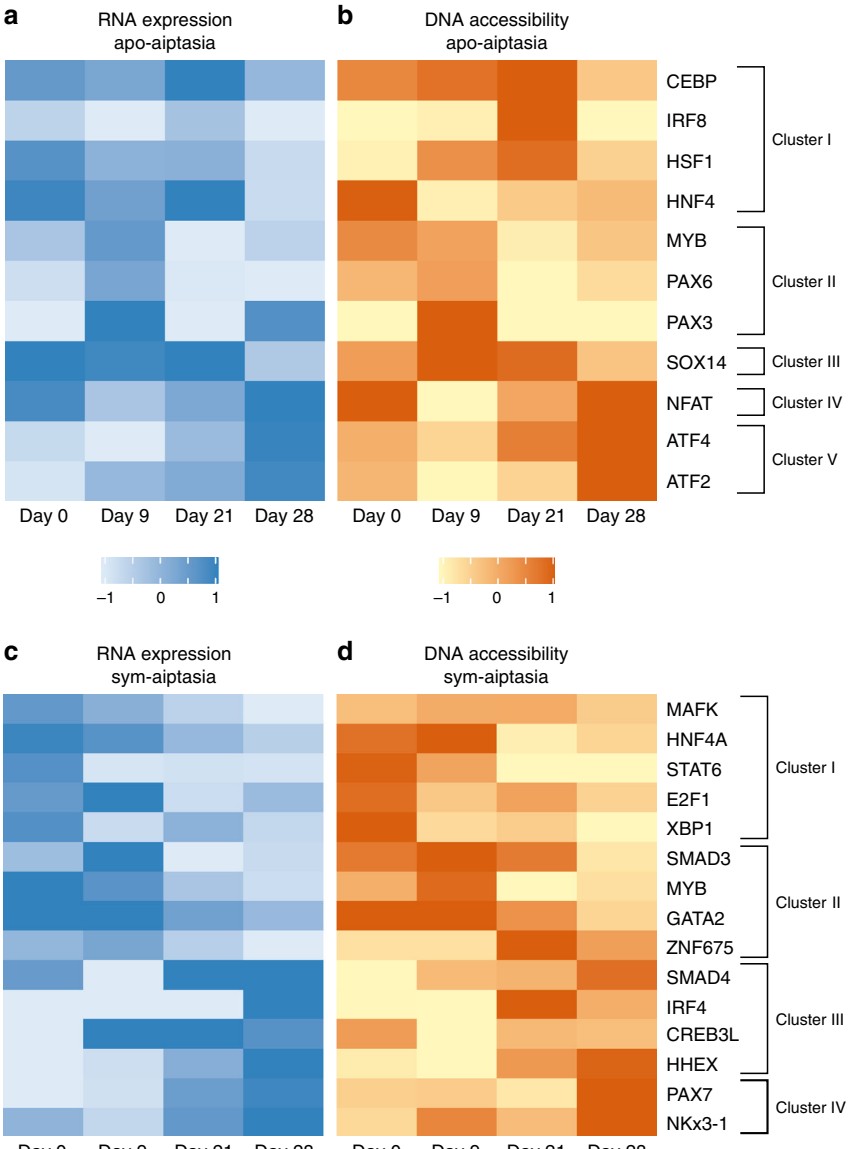

**Fig. 5** Transcription factors expression and accessibility found enriched within accessibility patterns. **a, b** Transcription factors found enriched within apo-symbiotic Aiptasia clusters. In red – (−log2(fold change)) of RNA gene expression. In green – (−log2(fold change)) of ATAC-seq accessible sites. R2 = 0.647, F (1,42) = 30.174, *P*-value < 0.01. **c, d** Transcription factors found enriched within symbiotic Aiptasia (symbiotic) clusters. In red – (−log2(fold change)) of RNA gene expression. In green – (−log2(fold change)) of ATAC-seq accessible sites. $R^2$ = 0.425, F (1,58) = 12.815, *P*-value < 0.01

observation is a result of altered transcription factor networks activated in response to the changing temperature and the presence (or absence) of the symbiont. Many cnidarians engage in complex symbiotic associations, comprising the eukaryotic host, photosynthetic algae, and highly diverse microbial communities; together referred to as holobiont[18]. The two symbiotic morphs of Aiptasia construct different holobiont relationships where the major symbiont of the symbiotic Aiptasia is the dinoflagellate *symbiodinium* and the apo-symbiotic Aiptasia has its bacterial and viral flora. This forces the host to respond differently and can directly affect the individual tolerance potential. Several processes, including DNA methylation, changes in histone density and variants, including various histone modifications and chromatin accessibility, have been described as regulators of various developmental and defense responses[30]. In the apo-symbiotic morph, the *NFAT* motif was enriched within cluster IV sites. There is evidence that the *NFAT* signaling pathway participates in the regulation of cell survival in different tissues and cell types[31].

Apo-symbiotic cluster V was enriched with *ATF4* motifs, this transcription factor is associated with homeostasis in response to ER stress, and showing higher levels when reaching 34 °C[32]. Altogether, transcription factor motifs that were found are also associated with various immune response pathways (*GATA3*[33], *SOX14*[34], *PAX3*[35]). In the symbiotic morph, the production of reactive oxygen species (ROS) is increased (Fig. 6c) in both the host and the symbiont cells in response to elevated temperatures[36]. This leads to the enrichment of motifs of *NKx3-1*, *SMAD*, *HNF4A* and *IRF4* transcription factors as detected by chromatin accessibility. These motifs are classified to act in response to oxidative stress in mammalian cells[37–39], as previously shown that *SMAD* protein family helps in improving mammalian cell response to oxidative stress[40]. Motifs that were found in both morphs are *MYB*, *CAD* and two representatives of the paired box (*PAX*) family are all important for cell homeostasis. Further analysis of enriched transcription factor motifs led to a review of their expression patterns. We identified a correlation between

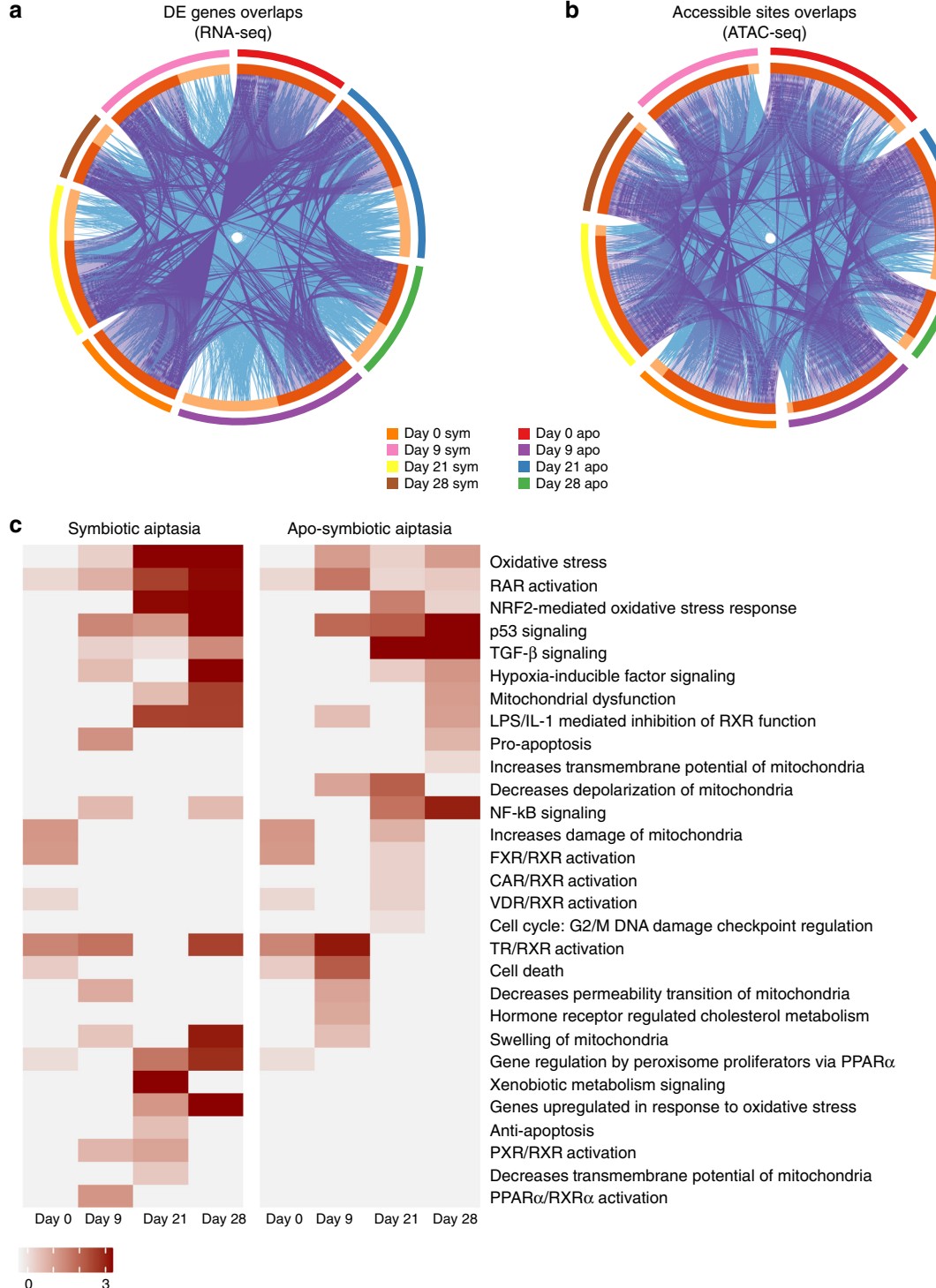

**Fig. 6** Canonical pathways enrichment analysis demonstrate different reactions to heat stress of morph. **a**, **b** Overlap among gene lists at the gene level, where purple curves link identical genes and blue curves link genes belong to the same enriched ontology term. The inner circle represents gene lists, where hits are arranged along the arc. Genes hit multiple lists are colored in dark orange, and genes unique to a list are shown in light orange. **c** Top canonical pathways found enriched within morphs were selected and compared to each other reviling different response to heat-stress

expression and accessibility of transcription factors found enriched in clusters. Moreover, we identified resemblance in their expression/accessibility to their relative cluster pattern showing a link between those TSs and the observed transcriptional heat-stress response.

The canonical pathways enrichment analysis using RNA-seq further strengthened the differences in response to heat we observed (Fig. 6). Among top canonical pathways, apo-symbiotic morphs were enriched with immunological response loci such as *TGF-β* Signaling, *NF-kB* Signaling, and p53 Signaling. In contrast, symbiotic sea anemones were more enriched with pathways related to oxidative stress showing gene upregulation including RAR activation, NRF2-mediated and other oxidative stress responses. We concluded that these changes are due to symbiont

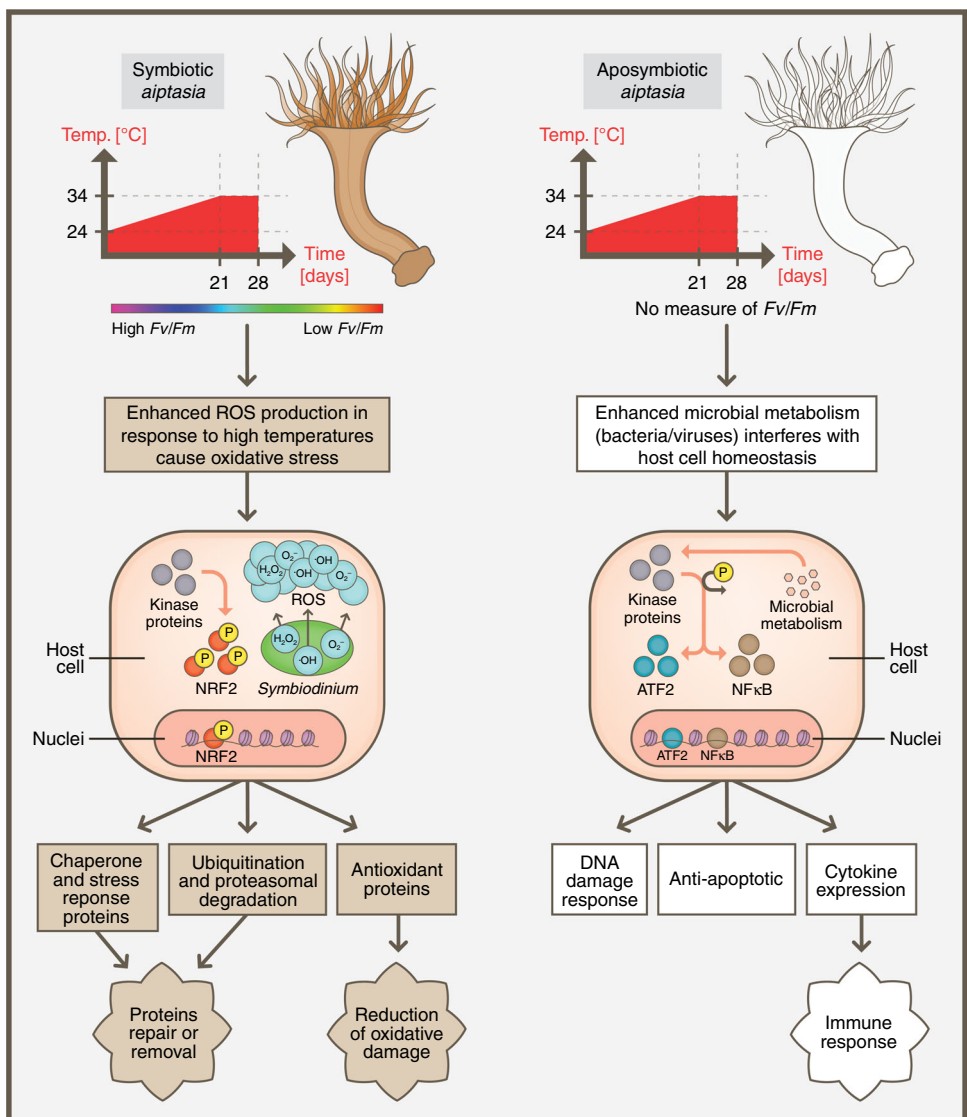

**Fig. 7** Illustration showing the different response pathways to heat-stress of two morphs of Aiptasia pallida – symbiotic and apo-symbiotic. The temperature elevation clearly initiates different secondary stressors within the sea anemone (host) cell. These triggers relevant sites in the genome to be more accessible to allow transcription factors regulation and gene expression affecting cell fate – death or survival

presence and, consequently, the elevation of ROS, while apo-symbiotic morphs responded to possible changes in microbial communities. Previously this phenomenon was observed before as a major cause leading to coral bleaching as reflected by the decline of *Fv/Fm* measurements of symbiotic Aiptasia (Fig. 1b)[41,42]. Finally, stress state triggered by high temperature is causing different response pathways. This is due to different secondary stress that is stimulated within the host cell depending on its symbiotic state. Many studies have shown that heat triggers increase ROS production by symbiont cells[43,44], and for symbiotic Aiptasia morph the response to it is crucial for cell survival. However, apo-symbiotic Aiptasia morphs are faced with increased microbial metabolism that can either help them adapt to heat or harm their health[18,45]. Based on chromatin dynamics and the gene expression landscape we have designed a scheme suggesting the molecular pathways occurring in response to heat-stress depending on Aiptasia symbiotic state (Fig. 7).

Finally, to our knowledge, this work presents the first analysis showing the relationship between gene expression and DNA accessibility by integrating two sequencing methods for a better understanding of the model organism, Aiptasia, regulatory

landscape. Thus, exposing the regulatory elements that participate in genome-wide gene regulation, in response to thermal stress and symbiont presence can serve as an important glance into the role of the symbiotic algae presence. Our findings are important for cnidarian and evolutionary investigations as well as a step towards deepening our understanding of cnidarian epigenetic adaptive mechanism to climate change in the marine milieu.

## Method

**Animal culture**. Adult *Aiptasia pallida* were kept in plastic containers filled with ten liters of artificial seawater at a salinity of 35 ppm, under natural light and a constant temperature of 24 °C or gradually changing temperature starting at 24 °C and max temp of 34 °C (temperature rose daily by 0.5 °C). 100 individuals were kept in each container in a closed water system and fed five times a week with freshly hatched brine shrimp (*Artemia nauplii*). To identify the clade present, DNA was extracted from symbiotic *Aiptasia pallida* using the DNeasy kit (Qiagene, Germany) and analyzed by RT-PCR, using a set of four clade-specific primers (A-D) following published protocols[46]. The clade was identified as Breviolum (i.e., clade B)[24]. Photosynthetic efficiencies were measured in anemones with Imaging-PAM (pulse amplitude modulation; Maxi-PAM, Walz Gmbh, Effeltrich, Germany). The resulting images were analyzed using the Imaging-Win software program (v2.00 m,Walz Gmbh, Effeltrich, Germany).

**ATAC-seq nuclear isolation and library preparation**. Nuclei were isolated from adult *Aiptasia pallida* that were subjected to different temperature treatments. From each sample, tissue was suspended in 500 μL PBS-NAC 2% (N-acetyl-cysteine, sigma) by pipetting in a 1.5 mL tube[47]. The suspension was centrifuged at $1500 \times g$ for 5 min at 4 °C. The pellet was re-suspended in 500 μL PBS, and cells were counted using a cytometer. To remove algae contamination, 400,000 Aiptasia cells were centrifuged at $600 \times g$ for 10 min and supernatant was collected, then re-suspended in 500 μL PBS and centrifuged at $1500 \times g$ for 5 min at 4 °C. The pellet was suspended in 50 μL of ATAC-seq lysis buffer (10 mM TRIS-Cl pH 7.4, 10 mM NaCl, 3 mM MgCl$_2$, 0.1% IGEPAL CA630) and centrifuged at $300 \times g$ for 10 min at 4 °C. The supernatant was collected and kept in a 1.5 mL tube on ice. The pellet was re-suspended in 50 μL ATAC-seq lysis buffer and centrifuged at $300 \times g$ for 10 min at 4 °C. The supernatant was combined with the supernatant from the previous step. Then 9 μL of isolated nuclei were stained with DAPI to verify the isolation of intact nuclei. The isolated nuclei were then centrifuged at $1500 \times g$ for 10 min at 4 °C. Immediately following this centrifugation step, the pellet was re-suspended in the transposase reaction mix (25 μL 2× TD buffer, 2.5 μL transposase (Illumina REF: 15028212) and 22.5 μL nuclease-free water). The transposition reaction was carried out for 30 min at 37 °C. Directly following transposition, the sample was purified using an Invitrogen PureLink PCR purification kit (REF: K310001). Following purification, library fragments were amplified using 1 × NEBnext PCR master mix (#M0541S) and 1.25 μM of custom Nextera PCR primers forward and reverse (Supplementary Table 2), using the following PCR conditions: 72 °C for 5 min; 98 °C for 30 s; and a variable number of cycles as needed (we added 4–9 cycles) at 98 °C for 10 s, 63 °C for 30 s and 72 °C for 1 min. To reduce GC and size bias in our PCR, we monitored the PCR reactions using qPCR to stop amplification before saturation. To do this, we amplified the full libraries for five cycles, after which we took a 4-μL aliquot of the PCR reaction and added 6 μl of the PCR cocktail with Sybr Green (Promega, REF: A6001) at a final concentration of 0.6 ×. We ran this reaction for 20 cycles to determine the additional number of cycles needed for the remaining 46-μl reaction. The libraries were purified using Agencourt AMPure XP beads (cat. No. 63881) and analyzed on a Tape-Station.

**RNA-seq libraries preparation**. Apo-symbiotic and symbiotic Aiptasia individuals were sampled for RNA analysis by snap freezing in liquid nitrogen and were immediately transferred to −80 °C for storage. RNA was prepared using a combined RNA extraction procedure including Trizol reagent and an RNeasy Mini Kit (Qiagen)[48]. Aiptasia tissue was homogenized in Trizol (1 mL per 0.1d) and incubated for 5 min at room temperature. Samples were centrifuged for 10 min at $12,000 \times g$. Chloroform was added to the supernatant (0.2 mL chloroform per 1mL Trizol). Samples were centrifuged at 10,000×g for 18 min. Then the aqueous phase was transferred with an equal volume of 100% RNA-free EtOH to RNeasy mini kit column. RNA concentrations were determined using a NanoDrop (ND-1000) spectrophotometer, and the integrity was assessed by Tape-Station.

RNA samples with integrity values (RINs) >9 were used for deep sequencing analyses. RNAseq libraries were pooled and prepared from 1.5-μg aliquots of RNA ($n = 10$ per pool) for each treatment and time point (total of 14 samples), using the Illumina NEB ultra II RNA Library Preparation Kit v2 kit, according to the manufacturer's protocol.

**Data analysis**. Sequence data of whole Aiptasia ATAC-seq libraries were prepared using single end 50 bp reads from a single-end (SE) Illumina HiSeq run. Treatments were run on one lane of Illumina HiSeq2000. Sequenced reads were aligned to the *Exaiptasia pallida* genome using bowtie[22]. Only unique mapped reads were used. Peaks were called by applying MACS2[49], with the following parameters: -g 260,000,000 −nomodel −extsize 75−shift -30. transcription factor binding motifs enrichment were identified within the peaks using scripts within HOMER[50]: findMotifsGenome.pl and annotatePeaks.pl were used with default parameters, and the genome used as background. The raw counts were normalized using DESeq2, and transcripts with a normalized value larger than 30 were retained. The sequencing data reported in this study has been deposited to the Sequence Read Archive (SRA) BioProject, under accession: PRJNA518019. To compare chromatin accessibility with expression patterns, we evaluated a set of ~6000 transcripts that were found to be differentially expressed. The genes had been sorted into five or four clusters with similar temporal expression patterns using a K-means clustering analysis.

Further analysis was performed using the BEDTools suites[51] (bedtools bamtobed, bedtools shift, bedtools merge, bedtools jaccard, bedtools intersect, bedtools window, bedtools getfasta) with default parameters. GO for biological processes and gene promoters found within treatment specific peaks were defined, and were subsequently analyzed for enriched terms using metascape.org[52] and Causal variants in this study were identified through the use of Ingenuity® Variant Analysis™ software https://www.qiagenbioinformatics.com/products/ingenuity-variant-analysis from QIAGEN, Inc.

**Statistics and reproducibility**. Physiology assay results were compared using SPSS one-way ANOVA with post-hoc Tukey-SDH. RNA-Seq and ATAC-seq results were analyzed using the R package DEseq2 (v 1.22.2) to detect statistically significantly

changed expression or accessibility respectively of genes/promoters, requiring a P-value of ≤0.05 and –log2(fold-change) <−0.6 or –log2(fold-change)>−0.6. heat maps of all the significantly accessible/expressed genes were generated using the heatmap.2 function from the R BIOCONDUCTOR package GPLOTS (v2.17.0). Heatmap.2 was used with the default clustering method and scaling the data by rows. GO for biological processes and gene promoters found within treatment specific peaks were defined, and were subsequently analyzed for enriched terms using metascape.org52 and Causal variants in this study were identified through the use of Ingenuity® Variant Analysis™ software https://www.qiagenbioinformatics.com/products/ingenuity-variant-analysis from QIAGEN, Inc.

**Reporting summary**. Further information on research design is available in the Nature Research Reporting Summary linked to this article.

## Data availability

The sequencing data reported in this study has been deposited to the Sequence Read Archive (SRA) BioProject, under accession: PRJNA518019. Peaks BED and excel files are available at figshare[53]. Correspondence and requests for materials should be addressed to the authors.

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

## Acknowledgements

The research leading to this paper has received funding from the Moore Foundation, "Unwinding the Circadian Clock in a Sea Anemone" (Grant #4598) to A.T & O.L. and funding from the Israeli Science Foundation (ISF), Grant #580/19 to O.L. We would like to thank Ms. Adi Zweifler and Dr. Noa Simon Blecher for their help with Aiptasia pallida cultures and assistance. We also would like to thank The Interuniversity Institute for Marine Sciences in Eilat (IUI) for the support in this research. This study represents partial fulfillment of the requirements for a Ph.D. thesis for E. Weizman at Faculty of Life Sciences Bar-Ilan University, Israel.

## Author contributions

E.W. carried out experiments and data analysis and wrote the paper. O.L. conceived the study and wrote the paper.

## Additional information

**Competing interests:** The authors declare no competing interests.

