## [Peer Review File · Communications Biology]

Reviewers' comments:

Reviewer #1 (Remarks to the Author):

In this work, the authors shed light on the chromatin regulation and gene expression of the sea anemone *Aiptasia pallida* (a cnidarian model widely used to study coral symbiosis) under conditions simulating the current situation of global ocean warming (i.e., comparing individuals cultured at a control temperature of 24 °C and individuals subject to a relatively rapid increase of water temperature, reaching 34 °C in 21 days). Taking advantage of the facultative symbiosis between *A. pallida* and the dinoflagellate *Symbiodinium*, they compare the response to thermal stress of symbiotic (sea anemone + *Symbiodinium*) and apo-symbiotic (sea anemone) individuals. They combine ATAC-seq and RNA-seq methodologies in order to investigate how the chromatin of these cnidarians is regulated during the response to heat stress and how the symbiotic state influences this response. Interestingly, they find striking differences between the two morphs regarding the chromatin accessible sites and gene expression. ATAC-seq experiments show enrichment on accessible chromatin motifs related to oxidative stress response in symbiotic individuals, and related to cell homeostasis and immune responses in the case of apo-symbiotic individuals. Interestingly, RNA-seq analysis reinforces those results showing similar enrichment of expressed genes involved in those pathways. Based on these results, they depict a plausible scenario where heat stress triggers ROS production in symbiotic individuals causing oxidative stress and a consequent cascade response to this damage; while heat stress in apo-symbiotic individuals causes metabolic changes in virus and bacteria of their microbiome triggering an immune response by the cnidarian host.

I did not find any major problem with this study. In my opinion, the paper is interesting for the community and brings relevant information on a critical topic that is of great concern these days. Although it is well-known that heat stress produces oxidative damage in symbiotic cnidarians (mainly in corals) due to ROS production, the authors provide valuable information on how the chromatin of these animals is regulated on these events, showing an interesting correlation between the expressed genes and the chromatin's accessible landscape. In this sense, one of the major points of this work is the successful application and fine-tuning of ATAC-seq technology to the cnidarian model *A. pallida* using whole animals. This technique should be an important tool for researchers in the field.

Overall, I do not have any major criticism: abstract and introduction sections are well written and provide enough information to understand the work, the experimental design was well performed, and the discussion and conclusions described in the manuscript are in agreement with the obtained results.

Methods are generally well explained and provide enough information to replicate the experiments. However, there are a few aspects I feel deserve clarification:

1-Please, specify how the measurements of the photosynthetic yields were performed (i.e., model of PAM fluorometer if that was the case).

2-There is no data in the SRA database under the accession PRJNA518019. This information should be made available before publication.

3-When using MACS2 for peak calling, why do the authors set the --extsize and --shift parameters in 75 and -30, respectively? I've seen most people using 73 and 37 accounting for the DNA size wrapped by the nucleosome.

4-Lines 105-107: "The experimental design included four groups, in which two were introduced to

heat-stress and two to constant temperature conditions serving as the control, over a period of 28 days. This was done for both symbiotic and apo-symbiotic state." Based on these sentences, it is not clear to me whether the authors included four groups for each morph or just four groups for the whole experiment as depicted in table 2.

5-Lines 304-331: Primer sequences should be presented in a more readable format (maybe as a table).

6-Lines 400-401: What further analysis had been performed with the BamTools and BEDTools suites?

Other minor typos I spotted:

-Line 182: "AND" should read "and"

-Line 214: "were" I think it should read "where"

-Line 281: ". then" should read ", then" or modify it accordingly.

-Line 285: "The pellet was re-suspended in 50 uL...." Please, indicate if you used ATAC-seq lysis buffer or some other buffer in this step.

-Figure 6C: Legends identifying the samples should be included (i.e, day 0, day 9, day 21, day 28, as in figure 5).

Reviewer #2 (Remarks to the Author):

In this manuscript, in order to expose the role of chromatin dynamics in response to thermal stress and how it is affected by the cnidarian (Aiptasia)-dinoflagellate symbiotic state, the authors carried out Transposase-Accessible Chromatin with high-throughput sequencing (ATAC-seq) and RNA-seq. They obtained samples of symbiotic and apo-symbiotic Aiptasia cultured over one month with 34°C. They identified 1309 genomic sites (853 in apo-symbiotic morph, 787 in symbiotic morph and 331 overlapping sites) that change their accessibility in response to thermal changes. Among transcription factors, the apo-symbiotic Aiptasia accessible sites are enriched with NFAT, ATF4, GATA3, SOX14 and PAX3 motifs and expressed genes related to immunological pathways. On the other hand, symbiotic Aiptasia accessible sites were enriched with NKx3-1, HNF4A, IRF4 motifs and expressed genes related to oxidative stress pathways. This study provides an information on molecular path towards understanding thermal stress gene regulation with the association of gene activity and histone modifications. Basically I recommend the publication of this manuscript but there are several concerns the authors could address for the acceptance.

Comments

(1) The term "coral" should be cautious or precisely used, since *Expaptasia pallida* (Aiptasia) is a sea anemone but not coral. Aiptasia may be a model system to study cellular and molecular mechanisms involved in cnidarian-dinoflagellate symbiosis, but it's not coral-dinoflagellate symbiosis.

(2) There is no detailed information of Symbiodinium. What clade was used in this experiment? Or recently LaJeunesse et al. (Curr Biol 28, 2570–2580, 2018) discussed systematic revision of

Symbiodiniaceae. The relationship of cnidarian-dinoflagellate symbiosis broadly changes depend on species of Symbiodiniaceae. Without the information of Symbiodiniaceae in the main text, one cannot repeat the experiment.

(3) Table 1 may be as Supplementary Information.

(4) lines 144-145: “---, as has been shown before (Sorek et al., 2018)- Figure 2 C-F.” should be revised.

(5) A weak point in this study is that they showed no data on gene expression changes that occur in symbiotic dinoflagellate itself. Since the difference in gene expression profiles between apo-symbiotic and symbiotic Aiptasia derived from mutual co-work between the host and symbiont, although they describe some in the discussion section and Figure 6, the authors should show what happened in the Symbiodinium side. Readers want to know the date to understand the real meaning of this study.

Reviewer #3 (Remarks to the Author):

This study looks at the role of aiptasia symbiosis in response to heat stress. The authors subject symbiotic aiptasia and apo-symbiotic animals to increased water temperatures over the course of 4 weeks. Animals were sampled several times over the course of the experiment for ATAC-seq and RNA-seq analysis, with downstream computational analysis to identify pathways involved in heat stress response. The main observation is that symbiotic animals respond to heat stress by activating redox pathways, while apo animals mount an immune response.

The strength of this study is the genomic analysis of the effects of stress on an interesting model organism that exhibits facultative symbiosis. The weakness is in the extent of some of the analysis and the level of experimental validation of genomic observations. Ultimately, while I can see the differences, I'm left wondering why the two symbiotic states respond differently to stress.

I have the following questions / comments that should be addressed before publication:

1) The authors imply several times (including the abstract) that their study sheds light on the role of histone modifications, when what their assay detects in chromatin accessibility. This language should be clarified.

2) The sample labeling throughout was confusing. I'm guessing that Control_sym_s1 is the Day 0 - 24C sample, etc, but the authors should use a clearer, systematic, labeling system. I suggest including the day of treatment, the temperature, and the symbiotic state.

3) I'd like to see bright field images of the animals with heat treatment. Do both symbiotic and apo animals show the same morphological phenotype? I can see that the symbiotic animals are undergoing morphological changes based on photosynthetic yield, and the authors state that both classes undergo morphological change, but I cannot tell what is happening morphologically to the apo animals.

4) Even without heat treatment, symbiotic and apo animals have differences in ATAC-seq profiles, shown in Fig. 2C and discussed by the authors on page 6. Have the authors explored these differences

to ask whether the differential response to heat treatment could be predicted based on the presence or absence of dinoflagellates? Does motif analysis of peaks unique to either symbiotic (n~50,000) or apo (n~15,000) animals give any insights into their stress response?

5) Related to point 3 above and the morphology of the apo animals – I find it interesting that the apo profiles from Day 21 / 34C differ so greatly from Day 28 / 34C, and specifically, that NFAT5 motifs are enriched in open regions that are unique to prolonged exposure to high temperature. This is in contrast to the symbiotic animals where many of the open regions seem similar between Day 21 and Day 28. Does this simply indicate that the apo animals are more stressed than the symbiotic animals, or that more of the animals are dying at a faster rate?

6) The columns in Fig. 6C should be labeled.

7) It would be nice to see some experimental validation of the genomic analysis. Are ROS levels actually increased in the symbiotic compared to apo?

8) Can the authors provide any insight on how the different relationships (dinoflagellate vs bacterial/viral flora) influence stress response? Can dinoflagellates be reintroduced to the apo animals mid-way through the experiment, and will that alter the stress response?

Reviewers' comments:

Reviewer #1

1-Please, specify how the measurements of the photosynthetic yields were performed (i.e., model of PAM fluorometer if that was the case).

L254-267: The measurements are added in the model of imaging pam mentioned under Method - Animal culture.

2-There is no data in the SRA database under the accession PRJNA518019. This information should be made available before publication.

We have made the data now available. The data is now under bioproject: PRJNA518019.

3-When using MACS2 for peak calling, why do the authors set the --extsize and --shift parameters in 75 and -30, respectively? I've seen most people using 73 and 37 accounting for the DNA size wrapped by the nucleosome.

The table below represent the different parameters suit to different organisms. This is the reason we choose these parameters. From our calibration protocol this parameter of – shift and –extsize produced the most accurate peak results.

We have used the extsize 75 as used in the original ATAC-seq paper (Buenrostro, J. D., Giresi, P. G., Zaba, L. C., Chang, H. Y., & Greenleaf, W. J. (2013). Transposition of native chromatin for fast and sensitive epigenomic profiling of open chromatin, DNA-binding proteins and nucleosome position. Nature methods, 10(12), 1213.) and tuned the size parameter to final result of -30.

	Shift	extsize	-q	--call-summits	pmid
Capsaspora	none	40	0.01	Yes	PMC4877666
Drosophlia	50	none	0.01	no	25679813
human	37	73	0.05	no	26977395
human	-100	200	0.01-0.05	no	

Encode3 Kundaje lab	-37	73		Broad	https://www.biostars.org/p/209592/
Encode Kundaje lab	-75	150		Broad	https://www.encodeproject.org/documents/c008d7bd-5d60-4a23-a833-67c5dfab006a/@_@_download/attachment/ATACSeqPipeline.pdf
Mouse	75	none	0.01	yes	27869820
human	-25	50	0.01	no	ftp://ftp.sanger.ac.uk/pub/resources/theses/ka8/chapter5.pdf
Mouse	-50	100		yes	26949250
human	-100	200		no	28193859
zebrafish	-100 or 75	200 or 150			https://books.google.co.il/books?id=ef5ZCgAAQBAJ&pg=PA404&lpg=PA404&dq=ATAC-seq+MACS2&source=bl&ots=bpJNM6dmrc&sig=wpzGVCmWmFbt3I0jKshhndkNBuQ&hl=iw&sa=X&ved=0ahUKEwj177f-9tPVAhXoKsAKHY89Clg4ChDoAQhiMAw#v=onepage&q=ATAC-seq%20MACS2&f=false

4-Lines 105-107: “The experimental design included four groups, in which two were introduced to heat-stress and two to constant temperature conditions serving as the control, over a period of 28 days. This was done for both symbiotic and apo-symbiotic state.” Based on these sentences, it is not clear to me whether the authors included four groups for each morph or just four groups for the whole experiment as depicted in table 2.

L91-92: We have revised the sentence. “The experimental design included a total of four groups...”. 1 – symbiotic control, 2 – symbiotic introduced to stress, 3 – aposymbiotic control, 4 – aposymbiotic introduced to stress.

5-Lines 304-331: Primer sequences should be presented in a more readable format (maybe as a table).

Please find primers table under “ATAC-seq nuclear isolation and library preparation” supplementary table 2

6-Lines 400-401: What further analysis had been performed with the BamTools and BEDTools suites?

L352-353: We have used several commands from these suits, mainly to manipulate file formats.

Bowtie (we mentioned before in the paragraph), bedtools bamtofastq, bedtools shift, bedtools merge, bedtools jaccard, bedtools intersect, bedtools window, bedtools getfasta.

Please find edited text in the manuscript.

Other minor typos I spotted:

-Line 182: “AND” should read “and”

-Line 214: “were” I think it should read “where”

-Line 281: “. then” should read “, then” or modify it accordingly.

-Line 285: “The pellet was re-suspended in 50 uL....” Please, indicate if you used ATAC-seq lysis buffer or some other buffer in this step.

-Figure 6C: Legends identifying the samples should be included (i.e, day 0, day 9, day 21, day 28, as in figure 5).

All typos were corrected as suggested in the revised version. Thank you for the corrections.

Reviewer #2

(1) The term “coral” should be cautious or precisely used, since *Expaptasia pallida* (Aiptasia) is a sea anemone but not coral. Aiptasia may be a model system to study cellular and molecular mechanisms involved in cnidarian-dinoflagellate symbiosis, but it's not coral-dinoflagellate symbiosis.

L11: We corrected the text as follows: “study the cnidarian-dinoflagellate model *Exaptasia pallida*...”

(2) There is no detailed information of Symbiodinium. What clade was used in this experiment? Or recently LaJeunesse et al. (Curr Biol 28, 2570–2580, 2018) discussed systematic revision of Symbiodiniaceae. The relationship of cnidarian-dinoflagellate symbiosis broadly changes depend on species of Symbiodiniaceae. Without the information of Symbiodiniaceae in the main text, one cannot repeat the experiment.

L94 and L251-254: The clade used is Breviolum (i.e. clade B). We added this into the text.

(3) Table 1 may be as Supplementary Information.

We agree. Table 1 is now presented as supplementary table 1.

(4) lines 144-145: “---, as has been shown before (Sorek et al., 2018)- Figure 2 C-F.” should be revised.

L131: We revised the text as follows: “as been shown before - Error! Reference source not found. C-F.”

(5) A weak point in this study is that they showed no data on gene expression changes that occur in symbiotic dinoflagellate itself. Since the difference in gene expression

profiles between apo-symbiotic and symbiotic Aiptasia derived from mutual co-work between the host and symbiont, although they describe some in the discussion section and Figure 6, the authors should show what happened in the Symbiodinium side. Readers want to know the data to understand the real meaning of this study.

Reviewer 2 is right!! however in this study we have focused on the host (i.e. Aiptasia) response to thermal stress. Although this aspect of the model system is interesting and should be addressed in the future, there are works that study the relationship between host and endosymbiont expression. We also used the method of ATAC-seq which requires available genome which is not present at the moment for this work. Beside all recent epigenetic paper including methylations were dealing only with the host. We believe this is out of the scope of this current work. We hope in the near future to also include some data in relation to the symbiotic algae. We all need to remember this requires much more money to invest also in sequencing for ATAC-seq and RNA-seq.

Reviewer #3

- 1) The authors imply several times (including the abstract) that their study sheds light on the role of histone modifications, when what their assay detects in chromatin accessibility. This language should be clarified.

We meant to emphasize the relationship between some histone modifications and their influence on chromatin accessibility.
We now changed the text conformingly.

L18-19: “Our work opens a new path towards understanding thermal stress gene regulation with the association of gene activity and chromatin accessibility...”.
L200-201: “Several processes, including DNA methylation, changes in histone density and variants, including various histone modifications and chromatin accessibility...”

- 2) The sample labeling throughout was confusing. I’m guessing that Control_sym_s1 is the Day 0 - 24C sample, etc, but the authors should use a clearer, systematic, labeling system. I suggest including the day of treatment, the temperature, and the symbiotic state.

We have now relabeled as suggested table 1 and figure 1.

- 3) I’d like to see bright field images of the animals with heat treatment. Do both symbiotic and apo animals show the same morphological phenotype? I can see that the symbiotic animals are undergoing morphological changes based on photosynthetic yield, and the authors state that both classes undergo morphological change, but I cannot tell what is happening morphologically to the apo animals.

Please find IR images of PAM sampling in Figure 1A.

- 4) Even without heat treatment, symbiotic and apo animals have differences in

ATAC-seq profiles, shown in Fig. 2C and discussed by the authors on page 6. Have the authors explored these differences to ask whether the differential response to heat treatment could be predicted based on the presence or absence of dinoflagellates? Does motif analysis of peaks unique to either symbiotic (n~50,000) or apo (n~15,000) animals give any insights into their stress response?

We have tried to explore this question in our analysis. However, our results were inconclusive regarding this specific question. We analyzed motifs enrichment with the other morph peaks as background for enrichment. We did find more pathways related to symbiosis establishment and maintenance in sym unique peaks, and found pathways related to immune system function within apo unique peaks. These pathways did not change significantly when thermal treatment was induced and we couldn't find a predictor to future response as measured in our experiment. Therefore, we choose to focus in this work on sites that were significantly changed in response to heat, and, we left this question to future work. We do think that establishing these differences between the aiptasia morphs and regulatory network differences in basal conditions is important, and obviously there is a lot more to explore.

5) Related to point 3 above and the morphology of the apo animals – I find it interesting that the apo profiles from Day 21 / 34C differ so greatly from Day 28 / 34C, and specifically, that NFAT5 motifs are enriched in open regions that are unique to prolonged exposure to high temperature. This is in contrast to the symbiotic animals where many of the open regions seem similar between Day 21 and Day 28. Does this simply indicate that the apo animals are more stressed than the symbiotic animals, or that more of the animals are dying at a faster rate?

From our observations we believe that the stress response pathways are different due to the presence or absence of the symbiotic algae. There are many evidences from corals that bleached animals are more stressed, and in many cases this is the “Swan song” of the colony before its death. It is important to remember that in most corals symbioses is known to be “obligatory” which means that without the presence of the algae the corals will die. Now in aiptasia the symbiosis relationship is “facultative” which means that the sea anemones will survive without the symbiotic algae. Therefore, we do not think this necessarily means that the animals are more stressed in that case is just that the response is different. Different response pathways trigger different molecular level cascades and we think this is the reason for these changes. While symbiotic animals have to deal with ROS removals (as shown in the results) apo animals have to deal with immunological stressors that might be more versatile.

6) The columns in Fig. 6C should be labeled.

We have corrected figure 6 and labeled.

7) It would be nice to see some experimental validation of the genomic analysis. Are ROS levels actually increased in the symbiotic compared to apo?

As mentioned by Referee #1 (Chromatin dynamics in marine organisms) "it is well-known that heat stress produces oxidative damage in symbiotic cnidarians (mainly in corals) due to ROS production..."

and as published before, for example:

1- Cellular mechanisms of Cnidarian bleaching: stress causes the collapse of symbiosis

2- Multi-omics analysis of thermal stress response in a zooxanthellate cnidarian reveals the importance of associating with thermotolerant symbionts

3- Thermal Stress Promotes Host Mitochondrial Degradation in Symbiotic Cnidarians: Are the Batteries of the Reef Going to Run Out?

4- Regulation of Apoptotic Mediators Reveals Dynamic Responses to Thermal Stress in the Reef Building Coral *Acropora millepora*

5- Photosynthetic symbioses in animals

and many more.

By today it is well known and almost "text book" that ROS are increasing in the presence of the algae and during heat stress. We hope you understand our position and will positively reconsider the necessity of including this validation in our manuscript, as it is time consuming and we are hoping to get this paper as soon as possible due to its novelty with the methods used here.

8) Can the authors provide any insight on how the different relationships (dinoflagellate vs bacterial/viral flora) influence stress response? Can dinoflagellates be reintroduced to the apo animals mid-way through the experiment, and will that alter the stress response?

You propose another experimental design of the system. When stressed (especially in thermal stress) *Aiptasia* start to lose its symbiotic algae, to re-induce the symbionts to the animals would mean going back to base conditions (temp. of 24C) for a while. This will alter the long stress we tried to mimic. Many manipulations can be done to study the influence of holobiont, such as different clades of symbiodinium. We choose to focus on the host under long continuant thermal stress in this work. Although this is an important question, we believe it should be done in a separate work in the future to come.

REVIEWERS' COMMENTS:

Reviewer #1 (Remarks to the Author):

I want to thank the authors for addressing all my comments. I think that the new version of the manuscript reads better than the previous one and I do not have any further criticism.

Just pay attention to the in-text citations that are in different styles in some cases (e.g., lines 24, 25 and 27 versus lines 34, 40 and 41)

And Lines 280-281: (Error! Reference source not found.)...supplementary Table 2 should be referenced here.

Reviewer #2 (Remarks to the Author):

The authors have revised the manuscript properly including answers to questions I asked at the first review.

Two points the authors should address at the final revision.

(1) When Fig. 3A (apo-symbiotic Aiptasia) is compared to Fig. 4A (symbiotic Aiptasia), not only changes in expression of given genes but also changes in a global pattern/profile of genes expression are evident. Although the authors categorized them into five (Fig. 3A) and four (Fig. 4A), the clusters themselves differed between the two. For example, genes in the cluster I of the former sampling were highly expressed at 24C but the expression was suppressed at higher temperature (Fig. 3A). In contrast, the cluster I of the latter sampling contained genes that were expressed at 24C, higher at 28C, and highest at 34C. Namely, even in the clusters I, the content of genes were different, making a global difference in the profile. If possible, the authors should make some discussion on this point.

(2) There are several typo errors: for example, line 107. Please check carefully and once again.

Reviewer #3 (Remarks to the Author):

The authors were asked by the editors specifically to address two of my comments - (1) the differences in gene regulation (ATAC-seq peaks) between apo and symbiotic animals under basal conditions, and (2) experimental validation of the genomic observations.

I'll focus on #2 first - the authors state that it's "text book" that ROS levels increase in symbiotic animals during heat stress as rationale for not demonstrating this themselves. That's a defensible position, but I do feel that some experimental validation would have improved the rigor of of this manuscript.

#1 - Differences in gene regulation under basal conditions. The authors state that they observe no differences that can explain the differential response to heat shock. I still think it would improve the scope of this manuscript to include and discuss differences in the basal state before discussing differential response to stress.

Reviewer #1 (Remarks to the Author):

I want to thank the authors for addressing all my comments. I think that the new version of the manuscript reads better than the previous one and I do not have any further criticism.

Just pay attention to the in-text citations that are in different styles in some cases (e.g., lines 24, 25 and 27 versus lines 34, 40 and 41)

And Lines 280-281: (Error! Reference source not found.)...supplementary Table 2 should be referenced here.

Thank you. We have corrected and reviewed the manuscript again.

Reviewer #2 (Remarks to the Author):

The authors have revised the manuscript properly including answers to questions I asked at the first review.

Two points the authors should address at the final revision.

(1) When Fig. 3A (apo-symbiotic Aiptasia) is compared to Fig. 4A (symbiotic Aiptasia), not only changes in expression of given genes but also changes in a global pattern/profile of genes expression are evident. Although the authors categorized them into five (Fig. 3A) and four (Fig. 4A), the clusters themselves differed between the two. For example, genes in the cluster I of the former sampling were highly expressed at 24C but the expression was suppressed at higher temperature (Fig. 3A). In contrast, the cluster I of the latter sampling contained genes that were expressed at 24C, higher at 28C, and highest at 34C. Namely, even in the clusters I, the content of genes were different, making a global difference in the profile. If possible, the authors should make some discussion on this point.

Indeed, the lists of heat responsive sites from the two morph are different with only 331 overlapping sites. This indicates an altered response profile. We have addressed this in the discussion. L206-211.

(2) There are several typo errors: for example, line 107. Please check carefully and once again.

Reviewer #3 (Remarks to the Author):

The authors were asked by the editors specifically to address two of my comments - (1) the differences in gene regulation (ATAC-seq peaks) between apo and symbiotic animals under basal conditions, and (2) experimental validation of the genomic observations.

I'll focus on #2 first - the authors state that it's "text book" that ROS levels increase in symbiotic animals during heat stress as rationale for not demonstrating this themselves. That's a defensible position, but I do feel that some experimental validation would have improved the rigor of of this manuscript.

The authors agree that measuring ROS can make our finding more satisfactory for the readers. However, as we state this now after more than 30 years of cnidarians biology, we consider this a known fact. Levy et.al have published a few studies on the ROS issue in the past. Another consideration is that we do not have left enough tissue from the original experiment to produce statistically meaningful results. To repeat this experiment is too much time consuming to validate a text book fact. However, we will consider this a point for future studies.

#1 - Differences in gene regulation under basal conditions. The authors state that they observe no differences that can explain the differential response to heat shock. I still think it would improve the scope of this manuscript to include and discuss differences in the basal state before discussing differential response to stress.

The basal state differences have been addressed in L133-141 and 198-201.